# DYNAMIC GRAPH REPRESENTATION LEARNING WITH FOURIER TEMPORAL STATE EMBEDDING

## ABSTRACT

Static graph representation learning has been applied in many tasks over the years thanks to the invention of unsupervised graph embedding methods and more recently, graph neural networks (GNNs). However, in many cases, we are to handle dynamic graphs where the structures of graphs and labels of the nodes are evolving steadily with time. This has posed a great challenge to existing methods in time and memory efficiency. In this work, we present a new method named Fourier Temporal State Embedding (FTSE) to address the temporal information in dynamic graph representation learning. FTSE offered time and memory-efficient solution through applying signal processing techniques to the temporal graph signals. We paired the Fourier Transform with an efficient edge network and provided a new prototype of modeling dynamic graph evolution with high precision. FTSE can also prevent the 'history explosion' that exists in sequential models. The empirical study shows that our proposed approach achieves significantly better performance than previous approaches on public datasets across multiple tasks.

## 1 INTRODUCTION

Graph Representation Learning learns the graphs with low-dimensional vectors at nodes and graphs level. (Perozzi et al., 2014; Tang et al., 2015; Wang et al., 2016; Cao et al., 2015; Ou et al., 2016) In recent years, deep neural networks (DNNs) are extended to process graphical data and have been utilized in a plethora of real-time cases. (Estrach et al., 2014; Duvenaud et al., 2015; Defferrard et al., 2016; Li et al., 2015; Gilmer et al., 2017; Kipf & Welling, 2016; Hamilton et al., 2017; Jin et al., 2017; Chen et al., 2018; Veličković et al., 2018; Gao & Ji, 2019). One of the most popular networks is Graph Convolution Neural Networks (GCNs) which originated from Spectral Graph Theory but developed into spatial-based varieties. GCNs are natural extensions of Convolution Neural Networks (CNNs) which has been widely studied and used in many applications in different fields of research.

Traditional GCNs have achieved commendable performance on static graphs. However, many applications involved dynamic graphs where nodes and edges evolved steadily over time. For example, a social network is updated on a day-to-day basis as people developed new friends. The dynamic graph represented users' evolving social relationships. In financial networks, transactions between nodes naturally adopt time-stamps as temporal features. Transactions of different nature may perform differently in a financial network where our main focus is to find malicious parties. Learning the evolving nature of graphs is an important task where we predict future graphical features and classify nodes based on their past behaviors.

Learning evolving graphs poses great challenges on traditional GCN as temporal features can not be easily incorporated into learning algorithms. The simple way of concatenating GCNs with RNNs is straight forward in handling dynamic graphs, but it suffered from many drawbacks. We can summarize them as three folds: **Firstly**, the embedding vector of each node is not static and will be evolving with time. Models need to be capable of capturing the evolving nature. **Secondly**, the memory and computation cost for batch training is huge to keep multiple graphs from different timesteps in the memory at the same time. **Finally**, the large number of timesteps within a single batch brings difficulties to high precision modeling.

There is also a focus on using Deep Neural Networks to generate the graph embedding recently (Trivedi et al., 2018; Pareja et al., 2020; Xu et al., 2020) as another direction compared with traditional unsupervised dynamic graph embedding approaches (Nguyen et al., 2018; Li et al., 2018;

Goyal et al., 2018; 2020). Existing methods normally utilize Sequential Models (e.g.Recurrent Neural Networks(RNNs)) to learn temporal features. However, as the graph is a non-linear data structure, sequential model-based approaches are memory-costly to train and evaluate with the information from the whole graph as input. Meanwhile, pure GCN approaches built for static graphs are inefficient in capturing evolving features. Some approaches combining GCNs with RNNs (Trivedi et al., 2018; Pareja et al., 2020) are costly to evaluate due to the high time complexity induced by repeated Graph Convolutions as well as high space complexity caused by a large number of network parameters. Meanwhile, the RNN-based method could only see through a fixed amount of history timesteps in training, which makes the prediction imprecise.

We introduced Fourier temporal state embedding (FTSE) to address the above problem. Instead of using sequential models to model the evolving nature of edges, we formalize the existence of edges as a signal, transforming the original embedding problem into signal processing. We also designed a simple and efficient Edge-Convolution Network structure for FTSE and compared the complexity of it with RNN based approaches. Our main technique is Discrete-Time Fourier Transform (DTFT for short), which transforms the discrete temporal signal into its continuous frequency domain. Therefore, we can embed history timesteps into a fixed-length vector, enlarging the receptive field in a single batch. Our empirical study shows that FTSE is an efficient method in modeling temporal graphs and a good approach to model the signal with high precision.

We summarize the contribution of this work as follows:

1. We proposed Fourier Temporal State Embedding (FTSE) to learn dynamic graph representation via transforming time-series signal into the frequency domain. FTSE directly modeled the harmonic component rather than timesteps. We also designed a simple but potent edge convolution network to model continuous-time dynamic graphs. FTSE is also the first GCN based approach capable of modeling continuous time dynamic graphs (CTDGs).

2. We studied the drawbacks of sequential-based methods in time and space complexity and justified that FTSE has much lower complexity and smaller parameter scale, making it a more efficient alternative to sequential-based approaches. This has also been proven with experiments.

3. Extensive empirical study showed that FTSE significantly outperforms previous methods in convergence speed as well as model performance in Link Prediction, Node/Edge Classification, achieving more than 10% improvement on some datasets.

## 2 RELATED WORK

Many static network embedding methods are proposed to map the nodes to low-dimensional vector representations while preserving network structure as well as node information. Both supervised and unsupervised techniques have been designed. Dynamic graph embedding methods are often extensions of their static counterparts. DANE (Roweis & Saul, 2000; Belkin & Niyogi, 2002) used a matrix factorization-based approach to generate static node embeddings from eigenvectors of graph Laplacian matrix. This work was extended by (Li et al., 2017) by updating eigenvectors from previous ones to generate dynamic node embeddings. Random-walk based approaches (Perozzi et al., 2014; Grover & Leskovec, 2016) used normalized inner products of node embedding to model the transition probabilities in random-walk. These two approaches are extended by CTDANE (Nguyen et al., 2018), which proposes to walk on the temporal order.

Deep learning approaches are also popular in this area thanks to the flourishing new models. DynGEM (Kamra et al., 2017) used an autoencoding approach which minimizes the reconstruction loss as well as the distance of connected nodes in the embedding space. The point process-based approach is also popular in dynamic knowledge graph modeling. KnowEvolve (Trivedi et al., 2018) and DyRep (Trivedi et al., 2018) model the occurrence of edges as a point-process and model the intensity function with neural networks. DynamicTriad (Zhou et al., 2018) focuses on the basic 'triad closure process' where a group of three vertices is developed from unconnected vertices. They proved that this process is fundamental in graph evolving thereby making their model able to predict network dynamics. HTNE (Zhou et al., 2018) used the Hawkes process with the attention mechanism to determine the influence of historical neighbors. Point process based approaches are especially good at event time prediction.

Another set of approaches comes from the combination of Graph Neural Networks (GNNs) with Sequential Networks (e.g. RNNs). GNNs are used to digest topological information and RNNs are used to handle dynamism. Graph Convolution Neural Networks (GCNs) is one of the most explored GNNs in this setting. GCRN (Seo et al., 2018) proposed two methods. The first one is doing GCN first and feed its output to an LSTM to capture dynamism. The second one put LSTM as an alternative for Fully Connected Layer in GCN and feed it with node features. STGCN (Yu et al., 2018) proposed ST-Conv block which was composed of a 1D convolution on the temporal dimension of node features followed by a Graph Convolution layer and another 1D convolution. STGCN is designed for spatio-temporal traffic data, which had a fixed graph structure but dynamic node embeddings. EvolveGCN (Pareja et al., 2020) combines the two models together to form EvolveGCN-Unit. They propose two versions of it. In the first one, the GCN parameters are hidden state of a recurrent architecture that takes node embeddings as input. In the second one, the GCN parameters are the input of recurrent architecture.

## 3 PROBLEM FORMULATION

The dynamic graph has two notable varieties as has been formally defined in (Kazemi & Goel, 2020). *Continuous-time dynamic graph* (CTDG) can be represented as a pair $(\mathcal{G}, \mathcal{O})$ where $\mathcal{G}$ is a static graph representing the initial state of a dynamic graph at time $t_0$ and $\mathcal{O}$ is a set of observations where each observation is a tuple of the form (*event type, event, timestep*) where an event can be a 1-step modification on the graph structure.(e.g. edge addition, edge deletion, node addition) At any point $t \leq t_0$, we can obtain the snapshot $\mathcal{G}^t$ by updating $\mathcal{G}$ sequentially according to the observations $\mathcal{O}$ occured before time $t$.

The *discrete time dynamic graph* (DTDG) is the sequence of snapshots coming from the dynamic graph sampled at fixed space. DTDG is defined as the set $\{\mathcal{G}^1, \mathcal{G}^2, ...G^T\}$ where $\mathcal{G}^t = (\mathcal{V}^t, \mathcal{E}^t)$ is the snapshot at time $t$. $\mathcal{V}^t$ is the set of nodes in $\mathcal{G}^t$, and $\mathcal{E}^t$ is the set of edges in $\mathcal{G}^t$.

DTDG may lose information compared to their CTDG counterparts since it only includes snapshots at constant intervals. Models designed for CTDG are generally applicable to DTDG but the inverse is not necessarily true. CTDG problem can be approximated by DTDGs by aggregating $\mathcal{G}$ within a constant time period to form the snapshot. This time period is called *granularity*, denoting the length of time within a single timestep. The smaller the granularity, the better the approximation to the CTDG. Existing methods predominately focus on the DTDG problem and make approximations to the respective CTDG problem. FTSE, based on Fourier Transform, is capable of modeling CTDGs which we detailed in section 4. For a more detailed discussion on the implementation of them, please refer to appendix B.

The prediction problem could also be categorized into *interpolation* and *extrapolation*. Suppose we have information of $\mathcal{G}$ in time period $[t_0, t_T]$. In *interpolation* problem, we make predictions on some time $t$ such that $t \in [t_0, t_T]$. It is also known as the *completion* problem as has been studied in (Li et al., 2017; Leblay & Chekol, 2018; Dasgupta et al., 2018; Goel et al., 2019). In *extrapolation* problem, our goal is to make predictions at time $t$ such that $t \geq t_T$. *Extrapolation* is a more challenging problem as it is trying to predict the future based on the past. In this work we focus on *extrapolation* problem.

In some cases, the new observation is streamed to the model at a fast rate that we cannot update the model in an on-line fashion. This is also called streaming scenario in (Kazemi & Goel, 2020). This concept is similar to that of inductive/transductive learning, where the difference lied in whether or not using the training set to do inference on the testing set. Compared with the non-streaming scenario where we are able to retrain the model once new data comes, this scenario poses greater challenges on the model's capacity to generalize.

## 4 METHOD

In this section we introduce the proposed FTSE. Existing approaches see the DTDG as graphs set with timestep and tried to sequentially learn the evolving pattern of graphs, which incurred great computation overhead. Besides, not much progress has been made in modeling CTDG as modeling a continuous function on temporal axis is not easy for neural architecture. In section 4.1, we introduce

the definition of DTFT and DFT. In section 4.2 and section 4.3, we formally define FTSE and the memory efficient neural network EGC, later in section 4.4 we discuss the time complexity of FTSE and compared it with sequential based approaches.

## 4.1 PRELIMINARIES

Fourier Temporal State Embedding is based on Fourier Transform on Discrete-Time Signals. We firstly review discrete-time Fourier transform (DTFT): (Oppenheim, 1999)

**Definition 4.1 (Discrete-time Fourier transform)** *The discrete-time Fourier transform of a discrete set of real or complex numbers $x_n$, for all integers n, is a Fourier series, which produces a periodic function of a frequency variable. When the frequency variable, $\omega$, has normalized units of sample, the period is $2\pi$, and the Fourier series is :*

$$X_{2\pi}(\omega) = \sum_{n=-\infty}^{\infty} x_n e^{-i\omega n} \tag{1}$$

Since the result of DTFT is a continuous function with periodicity, we can compute an arbitrary number of samples $N$ in one cycle of the periodic function, which creates Discrete Fourier Transform (DFT). (Weinstein & Ebert, 1971; Briggs & Henson, 1995) Fast Fourier transform algorithm(FFT) is designed to speedily calculate DFT by factorizing the DFT matrix into a product of sparse factors. (Van Loan, 1992; Nussbaumer, 1981). Fast Fourier transform reduce the complexity of calculating DFT from $O(N^2)$ to $O(NlogN)$ where $N$ is the data size.

## 4.2 FOURIER TEMPORAL STATE EMBEDDING

In Fourier Temporal State Embedding (FTSE) we propose to formluate the existence of edges within a certain period of time (i.e. look back) as an aperiodic signal $f$ where

$$f(i,j,t) = \mathbf{Pr}(\text{node } i, j \text{ is connected at time } t) \tag{2}$$

Suppose we use information from $T$ historical timesteps $t_0, t_1, ...t_{T-1}$ we are able to use DTFT get the frequency response $F$ as

$$F(i,j,\omega) = \sum_{n=0}^{T-1} f(i,j,t_n)e^{-i\omega t_n} \tag{3}$$

which is a $w\pi$ period function of $\omega$. Then we can compute $N$ samples at interval $\frac{2\pi}{N}$ within a single period. This creates a $N$-length Fourier series $X$ where

$$\boldsymbol{X}_{i,j,k} = F(i,j,\frac{2\pi k}{N}) = \sum_{n=0}^{T-1} f(i,j,t_n)e^{-2\pi i \frac{kt_n}{N}} \tag{4}$$

This corresponds to DFT on the original temporal sequence when we have $N = T$. We thus embed the original temporal signal into its frequency representation at length $N$. This transformed the original temporal relation to the amplitudes and phases of its harmonic components.

Since we cannot take all history steps into one batch in training the model, the look back parameter $l$ is delimited by the maximum history timesteps the memory can hold. Our modeled function became

$$f_{target}(i,j) = \mathbf{Pr}(i,j \text{ is connected at time } t_T | f(:,:,t_i), \text{ where } i \in \{t_{T-l}, ..., t_{T-1}\}) \tag{5}$$

Where $f(:,:,t_i)$ denotes the all structural information in previous $l$ timesteps. However, as FTSE had a much larger receptive field, we can transform any number of timestamps into a $l$-dimensional vector by sampling $l$ values within a period in the frequency domain, achieving the goal of embedding a high volume of data into low dimensional representation. On the other hand, we remove the inner dependency between different timesteps and transform the time series learning into a new frequency learning problem. This makes the use of RNNs unnecessary. We did an empirical study in section 5.6.

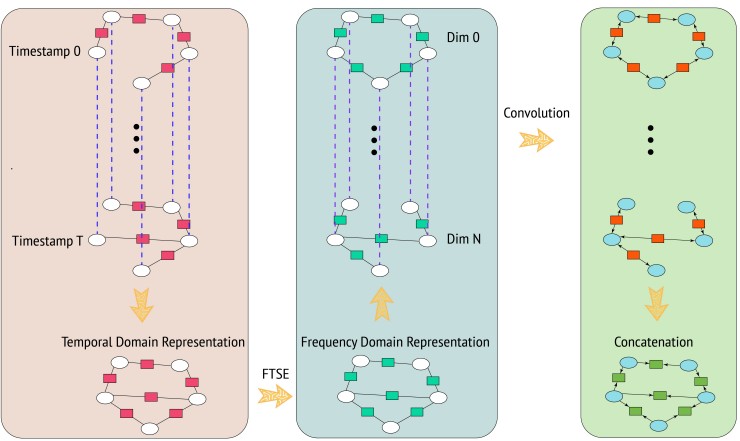

Figure 1: Work Flow of FTSE

### 4.3 NETWORK STRUCTURE

We designed a time-efficient, parallelizable Edge Convolution Neural Network called EGC suitable for FTSE to perform edge convolution on embedding vectors. EGC can handle edge-feature. There have been many approaches (Kim et al., 2019; Gong & Cheng, 2019) trying to incorporate Edge features into GCNs. EGNNs introduced the convolution on edges and update edge features in each layer together with the node feature. Attention-based approaches incorporate edge feature in attention score, which is determined by the significance of correlation between nodes (Xu et al., 2020). However, in our problem setting, multiple timesteps of an evolving graph are fed into the model in one pass. This poses great challenge to the memory if the model is overly complicated. Hence, we simplify the edge convolution with Vanilla GCN (Kipf & Welling, 2016) and incorporate the edge embedding naturally into the network structure. We did running-time comparison in the appendix 5, which justified the rationale of our design.

As has been discussed in 4.2, the $N$-dimensional FTSE is independent of each other, we perform $N$ independent graph convolutions based on the normalized adjacency matrix $\hat{A}^m$ (i.e. $\hat{A}^m = D^{-\frac{1}{2}}A^m D^{-\frac{1}{2}}$) where $A^m$ is the adjacency matrix replacing its entries with corresponding edge feature at entry $m$ (i.e. $A_{ij}^m = X_{ijm}$) and $D = diag(d)$ for $d(i)$ the degree of node i. Layer-wise forward propagation can be formulated as:

$$H_{t+1} = \sigma(\hat{A}^m H_t W_t) \text{ where i } \in \{0, 1, ..., N-1\} \tag{6}$$

In the above expression, $H_t$ is the hidden state of layer $t$, $W_t$ is the weight and $t$ is the index of current layer. We concatenate the embedding of graphs on different dimensions to form the final embedding. Finally, the edge embedding is calculated by concatenating the node embedding connected to it.

### 4.4 COMPLEXITY ANALYSIS

We compare the classic RNN+GCN approach with our proposed approach in time complexity. More complicated RNN structure (e.g. GRU , LSTM) and GCN structure (e.g. GAT, GraphSage) has even higher time complexity in training. We assume that GCNs are calculated with dense vector multiplication. For detailed analysis, please refer to the AppendixC.1.

**Time Complexity** In our proposed method, the time complexity is $O(|\mathcal{E}|T log T$ (FTSE) $+ |\mathcal{V}|^4$ (GCN) ) for forward propogation, assuming that we do graph convolution only once and no optimization on the matrix multiplication. The time complexity of RNN + GCN sturcture is $O(T \cdot (|\mathcal{V}|^2$ (RNN) $+ |\mathcal{V}|^4$ (GCN) )). The analysis shows that our algorithm compares favorably to RNN+GCN strcture in time complexity.

**Space Complexity** Assuming the GCN only had a single layer. For FTSE, the parameters take up $O(|\mathcal{V}|^2$ (GCN's) $+ N$ (MLP's) ) parameters. On the other hand, sequential-model based approaches

(the simplest RNN version) can take up $O(|\mathcal{V}|^2$ (GCN's) $+ N$ (MLP's) $+ T|\mathcal{V}|^2$ (RNN's) ) parameters, which has significantly more parameters than FTSE.

## 5 EXPERIMENTS

In the experiments section, we aim to answer the following questions: (1) How does FTSE perform on modeling dynamic graphs as compared to previous GCN+RNN based approaches? (2) What is the strength and weakness of FTSE? (3) How does each part of FTSE work? (4) How does FTSE handle the CTDG problem?

We evaluated and compared FTSE with state-of-the-art approaches on link prediction, node classification, and edge classification tasks on a variety of datasets with ablation study and running time experiments. Due to space limitations, we put some figures and extensive discussions in the appendix.

### 5.1 TASKS

We perform link prediction, edge classification, and node classification in the experiments: **Link Prediction** is a binary classification predicting the structure of an evolving graph at a certain time step $t$ based on existing information from time $\{0, 1, ..., t-1\}$. The classification is imbalanced with existing edges greatly outnumbered by non-exist edges, we apply 1000 times negative sampling and used weighted binary cross entropy as its loss function during training. In the validation and test phase, we used the whole graph. **Edge Classification** is the generalized link prediction task, where the edge has labels. We applied similar settings as link prediction class.**Node Classification** is to classify future node categorization. In three tasks, we used Average Precision (**AP**) as well as ROC-AUC score (**AUC**) as metrics to evaluate the final performance of the model.

In **streaming scenario** we perform inductive learning where the training set was not used as history timesteps in inference whereas transductive learning was carried out in non-streaming 3

### 5.2 DATASETS

We used public datasets from SNAP (Leskovec & Krevl, 2014) and previous work for evaluation.

**Autonomous System**[1](AS for short) Autonomous System (AS) is coming from (Leskovec et al., 2005) which is a communication network of routers. The edge represented the message exchange between routers and we used it to forecast the message exchange in the future.**Reddit Hyperlink Network**[2](Reddit for short) Reddit is a subreddit-to-subreddit hyperlink network where the hyperlink is coming from a post in the source community and links to another post in the target community. The hyperlink is annotated with timestep as well as sentiment, which makes it suitable for the classification of future edges.**UC Irvine messages**[3](UCI for short) UCI is a social network dataset creating out of students' message from the University of California, Irvine. The links within the dataset indicate the sent message between users. **Stochastic Block Model** (SBM for short) SBM is a random graph model for simulating graph structure and evolutions. The dataset comes from (Goyal et al., 2018). We did link prediction on this dataset. **Elliptic** Elliptic is a network of bitcoin transactions coming from (Weber et al., 2019). Each node in the graph represents one transaction and the edges indicate payment flows. Edges are labeled with the illicit and licit category. The goal is the categorization of unlabeled transactions.

The statistics of datasets are summarized in Table 2. The ratio is the negative vs the positive. Timesteps are divided with granularity following (Pareja et al., 2020; Goyal et al., 2018).

---

[1]http://snap.stanford.edu/data/as-733.html
[2]http://snap.stanford.edu/data/soc-RedditHyperlinks.html
[3]http://konect.uni-koblenz.de/networks/opsahl-ucsocial

## 5.3 BASELINES

We compared our proposed method with the following baselines based on deep learning approaches. **GCN**[4] is vanilla GCN without temporal modeling. The training loss is accumulated along timesteps. **GCN+GRU/LSTM** is a GCN model co-trained with GRU/LSTM model. The GRU/LSTM is fed with GCN's output in each timestep. This method is similar to the one in (Goyal et al., 2020). **EvolveGCN**[5] is proposed in (Pareja et al., 2020) This method is concatenating GCN with RNN to perform a new temporal graph convolution. **GAT**[6] is the graph attention networks proposed by (Veličković et al., 2018) . We also combine it with sequential models. **TGAT** is proposed in (Xu et al., 2020) using the temporal kernel to define the inner product in temporal attention and perform spatial-temporal convolution similar to the structure in (Hamilton et al., 2017). Another comparison was between the **EGNN**[7] (Kim et al., 2019) and our proposed edge convolution networks (**EGC** for short) at appendix D. We also implemented an easy version **TSE** where we didn't apply Fourier transform and use the direct temporal signal as embedding vector. The code is posted online.[8] All baselines are implemented in Pytorch.

## 5.4 EXPERIMENTAL SETTINGS

All experiments are performed on a Ubuntu 18.04 server with Nvidia Tesla V100 GPU (16GB memory) and Intel(R) Xeon(R) E5-2690 CPU. We did experiments 20 times for each model. Mean and standard deviation of AUC scores as well as mean score of AP are reported. The test result is based on the best performance epoch on validation set. Running time for every epoch as well as the best valid epoch number is also reported in the appendix. The experimental settings are identical for different model except the model-specific parameters.

| Datasets | UCI | | | AS | | | SBM | | |
|---|---|---|---|---|---|---|---|---|---|
| | **AUC** | **AP** | **Epochs** | **AUC** | **AP** | **Epochs** | **AUC** | **AP** | **Epochs** |
| GCN | 0.476(0.045) | 61 | 32 | 0.673(0.032) | 323 | 96 | 0.715(0.007) | 16587 | 7 |
| GCN+GRU | 0.509(0.027) | 260 | 64 | 0.825(0.013) | 935 | 9 | 0.718(0.004) | 17061 | 50 |
| GCN+LSTM | 0.512(0.023) | 383 | 56 | 0.801(0.017) | 1749 | 10 | 0.714(0.003) | **17954** | 57 |
| GAT | 0.592(0.023) | 284 | 423 | 0.876(0.023) | 1166 | 67 | 0.719(0.015) | 15843 | 10 |
| GAT+GRU | 0.508(0.007) | 227 | 14 | 0.658(0.030) | 313.2 | 24 | 0.719(0.008) | 16084 | 15 |
| GAT+LSTM | 0.511(0.004) | 316 | 17 | 0.728(0.029) | 424 | 18 | 0.718(0.009) | 15845 | 16 |
| EvolveGCN-O | 0.580(0.019) | 676 | 483 | 0.907(0.007) | **4636** | 82 | 0.717(0.005) | 16239 | 24 |
| EvolveGCN-H | 0.583(0.018) | 702 | 372 | 0.904(0.012) | **4906** | 106 | 0.715(0.003) | 16948 | 34 |
| TGAT | - | - | - | - | - | - | - | - | - |
| TSE | **0.718**(0.011) | **761** | 108 | **0.989**(0.002) | 2983 | 26 | **0.721**(0.008) | 17739 | 13 |
| Fourier TSE | **0.714**(0.018) | **728** | 115 | **0.986**(0.003) | 3012 | 24 | **0.723**(0.006) | **17817** | 12 |

Table 1: The ROC-AUC score (**AUC**) and Average Precison(**AP** at $1e-6$ scale) on **non-streaming** link prediction task, the mean and standard deviation of AUC score and the mean values of AP (due to space limitation) are reported according to 20 runs of each algorithm

## 5.5 RESULT AND DISCUSSION

**Link Prediction** In link prediction task, we experimented with 9 different baselines, the results are in 1 and 4. In the three datasets we benchmarked, the experimental settings are identical for all the baselines except the GNN parts. **TGAT** was not able to run normally on our devices and overflows the memory because it has the "neighbor explosion" problem. **EvolveGCN-O/H**'s model was computed in CPU due to the memory limit of GPU while others are fully computed with GPUs. Our proposed methods **TSE** and **FTSE** achieve the best result in AUC score on all 3 datasets. In

---

[4]https://github.com/tkipf/gcn
[5]https://github.com/IBM/EvolveGCN
[6]https://github.com/PetarV-/GAT
[7]https://github.com/khy0809/fewshot-egnn
[8]https://github.com/anonym-code/ICLR2021

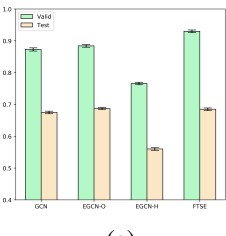 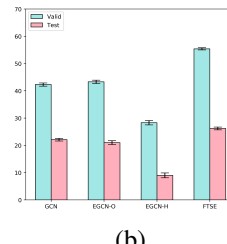 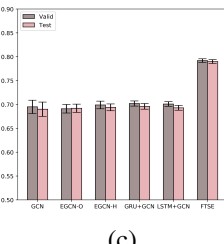 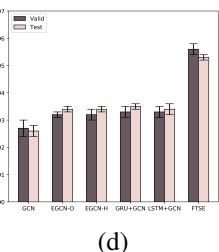

| (a) | (b) | (c) | (d) |

Figure 2: **(a)** ROC-AUC score on Node Classification task of Elliptic dataset **(b)** Average Precision on Node Classification task of Elliptic dataset **(c)** ROC-AUC score on Edge Classification task of Reddit dataset **(d)** Average Precision on Edge Classification task of Reddit dataset. Mean and standard deviation is reported based on 20 runs of algorithms

the average precision benchmark(**AP**), many previous approaches are still achieving commendable results. We also achieve moderate convergence speed compared with previous work.

**Edge Classification** In edge classification task, we compared FTSE with 5 baselines. The results are shown in Figure 2 The GAT based varieties are causing memory overflow and was not included. The result shows that FTSE outperformed the state-of-the-art by 12% in the AUC score and 3% in average precision.

**Node Classification** In the node classification task, we compared FTSE with vanilla GCN and two EvolveGCN varieties. The result shows that we outperformed state-of-the-art in the AUC score and AP. However, the performance on the validation set is more significant. We believe it is due to the fact that the validation set is predicted in a transduction setting where some of its previous timesteps have already been seen in the training set.

The result of those tasks justifies that FTSE/TSE is the new state-of-the-art method in modeling Dynamic Graph. It is especially capable of capturing edge link evolving. The result shows that we can achieve very well performance even without Fourier Transform. In our ablation study, we justify the rationale of the Fourier Transform.

### 5.6 Ablation Study

In the ablation study, we simulate the effect of CTDG via granularizing the timesteps into smaller intervals. We detailed its reasoning in Appendix B. We take on different values of look back parameters and embed the temporal state into a $k$ dimensional vector with **FTSE**. In **TSE**, we remove the compression process and use the last $k$ timesteps of the original temporal signal as edge embedding. We compared the streaming as well as non-streaming scenarios. The result in E shows that with increased compression ratio (i.e. increasing the look back) the performance of FTSE compares favorably to original TSE. This justifies that FTSE could handle cases where the granularity is small and performs a more accurate inference with the same computation and memory budget. Extensive discussion and empirical study of **FTSE** vs **TSE** is presented the Appendix.E

### 6 Conclusion

Graph Neural Networks have already been developed into many popular branches and justified its effectiveness in modeling graphical data. However, dynamic graph modeling is still an useful but difficult task awaiting future study. The straightforward GNN + RNN model is a bringing high computation overhead and memory cost, attention-based approach can induce high computation cost since the neighbors are spatial-temporal. Instead, we explored another branch where we see the state of edges existence as a signal in the temporal domain. This facilitates the use of signal processing techniques like DTFT to provide better quality frequency-domain embeddings with high precision. Our approaches achieved impressive performance over dynamic graph modeling in different scenarios confirmed by empirical study. Future direction of this work included merging GCNs with Temporal Signals and extensive study on the harmonic properties of graph signals.

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

## A   DATASET STATISTICS

| Datasets | Nodes | Edges | Class | Ratio | Timesteps (Train/Val/Test) |
|----------|-------|-------|-------|-------|----------------------------|
| SBM | 1,000 | 4,870,863 | 2 | 513: 1 | 35 / 5 / 10 |
| UCI | 1,899 | 59,835 | 2 | 5,216: 1 | 62 / 9 / 17 |
| AS | 6,474 | 13,785 | 2 | 30,390: 1 | 70 / 10 / 20 |
| Reddit | 55,863 | 858,490 | 2 | 1,489: 1 | 122 / 18 / 34 |
| Elliptic | 203,769 | 1,352,694 | 2 | 9,245: 1 | 31 / 5 / 13 |

Table 2: Dataset Statistics

## B   DISCUSSION ON CTDG VS DTDG

In this section, we discuss the actual implementation of the CTDG problem. Although CTDG is defined in the continuous temporal domain, it can always be approximated at any precision with a DTDG since we can granularize the time period into discrete timesteps of any length. The granularity $g$ depends on the number of timesteps we expected. For example, if we have a 1-year record of an evolving social network, we can split it into 365 days. We can aggregate all the connection information for each day and form 365 timesteps. This formulates the original CTDG into DTDG via taking snapshots over a small period (i.e. 1 day). Smaller granularity increased the prediction precision on the time axis (i.e. from the probability of having a connection in 1 day to 1 hour) but incurred more memory overhead when we are trying to take in the same historical period of data. Fourier TSE is better than TSE in that it transforms the original temporal signal into its frequency domain and generates a fixed-length embedding vector completely made of the harmonic component of the signal. We justify through ablation study that this is a rational approach to deal with low-granularity DTDG, yielding convincing performance.

## C  DETAILED COMPLEXITY ANALYSIS

First, we formalize the notation in this section. A graph $\mathcal{G}(\mathcal{V}, \mathcal{E})$ consists of $|\mathcal{V}|$ nodes and $|\mathcal{E}|$ edges, the batch sizes (number of timesteps) is $T$ and the dimension of embedded vector is $N$. Other parameters (i.e. hyperparameters like hidden state dimension) are treated as constant.

### C.1  TIME COMPLEXITY

**GCN** 1 layer Vanilla GCN with dense matrix multiplication is calculated as:

$$\boldsymbol{H} = \sigma(\hat{\boldsymbol{A}} \boldsymbol{W} \boldsymbol{X})$$

where $\hat{\boldsymbol{A}}$ is the normalized adjacency matrix (i.e., $\hat{\boldsymbol{A}} = \boldsymbol{D}^{-\frac{1}{2}} \boldsymbol{A} \boldsymbol{D}^{-\frac{1}{2}}$, where $\boldsymbol{D}$ is the diagonal degree matrix) $\boldsymbol{W}$ is the weight and $X_0$ is the original node feature. Since the matrix multiplication can be calculated in $O(N^3)$( although the upper bound can be reduced to $O(N^{2.3737})$ via Coppersmith-Winograd algorithm, we use the dumb version of it for simplicity) Then the complexity of performing a 1-layer graph convolution is $O(N^4)$ on the assumption that the hidden state dimension does not have the same scale as $N$. Hence the complexity in Graph Convolution is $O(N^4|T|)$ in total following that it perform convolution in every timestep.

**RNN** The simplest structure of sequential model is Simple Recurrent Neural Network (SRN), which is formulated as:

$$h_t = \sigma_h(\boldsymbol{W}_h x_t + \boldsymbol{U}_h y_{t-1} + b_h)$$
$$y_t = \sigma_y(\boldsymbol{W}_y h_t + b_y)$$

(There is also Elman network but they are similar in analysis.) The forward propogation has complexity of $O(|\mathcal{V}|^2)$ since $x_t$ is the adjacency matrix of Graph and has space complexity of $O(|\mathcal{V}|^2)$. Therefore, the RNN has computation complexity of $O(|\mathcal{V}|^2|T|)$ as a whole.

Thus, we conclude that the sequential-based model has forward complexity of $O(|\mathcal{V}|^4 T)$. Our algorithm, on the other hand, is parallelisable since it has no dependency between time steps. The FFT algorithm can be calculated in $O(|\mathcal{E}|T log T)$ presented This makes the total complexity $O(|\mathcal{V}|^4 + |\mathcal{E}|T log T)$.

### C.2  SPACE COMPLEXITY

The space complexity of neural networks is estimated from two parts: (1) model parameters (2) data volume.

1. In sequential-based models, the GCN part takes up $O(|\mathcal{V}|)$ parameters and RNN takes up $O(|\mathcal{V}|^2 T)$ since $x_t$ and $h_t$ are both $O(|\mathcal{V}|^2)$ length and there are $T$ units. This makes the total parameter complexity amount to $O(|\mathcal{V}|^2 T)$ in total. In our proposed approach, the edge-convolution model takes up $O(|\mathcal{V}|N)$ parameters.

2. sequential-based models take up $(O|\mathcal{V}|^2 T)$ active memory to store its data while FTSE takes up $(O|\mathcal{V}|^2 N)$.

To summarize, sequential-based models had $O(|\mathcal{V}|^2 T)$ in space complexity whereas our proposed approach takes up $O(|\mathcal{V}|^2 N)$ memory in total. In both parts, the sequential-model was outperformed by ours.

## D  RUNNING TIME EXPERIMENTS

We did 2 experiments about time complexity analysis.

**Edge Convolution Network** The first one compares EGNN(C) (Gong & Cheng, 2019) with our proposed EGC network on FTSE, result shows that EGC could significantly improve the training speed of networks with the same hardware platform. The result for training is more impressive than validation and test phase, which can be explained by the difference of their settings. In training phase we sampled a certain amount of 'non-exist' edges whereas in validation and test set the whole graph is evaluated.

| Datasets | UCI | | | AS | | |
|---|---|---|---|---|---|---|
| | Train | Valid | Test | Train | Valid | Test |
| FTSE+EGNN | 35.12(2.13) | 21.31(0.42) | 37.40(0.82) | 102.35(12.66) | 356.24(8.48) | 691.93(16.40) |
| FTSE+EGC | 26.68(1.43) | 19.27(0.40) | 33.95(0.67) | 80.16(6.28) | 345.18(8.00) | 646.24(12.78) |

Table 3: Training time of different GCN structures on link prediction task in seconds

**Time Complexity** We did an experiment justifying the time complexity of our proposed network. We used the exact same experimental setting for the two baselines and did experiment with the identical hardware platform. We test the performance on three link-prediction datasets. Results show that our method could shorten the computation time of RNN+GCN based approach by 50% on average, which corresponds to the previous complexity analysis.

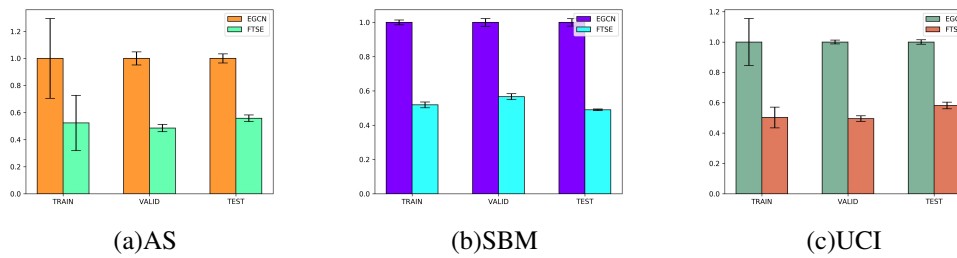

(a)AS          (b)SBM          (c)UCI

Figure 3: Running Time Result on FTSE and EGCN on three different datasets, the scale is on the running time of EGCN. FTSE's time and the standard deviation is adjusted correspondingly, the result is based on 20 runs of the algorithm under identical setting

# E  FTSE VS TSE

We did experiments comparing FTSE and TSE. The goal is to learn the dynamic graph with high precision, where we need to use a huge amount of history timesteps (i.e. look back is very high) but could only use fixed-dimension embedding. Both FTSE and TSE are using the same number of embedding dimensions for the time series and the network is EGCN. The result showed that FTSE outperforms TSE when the compression ratio is high, justifying its capability of modeling a dynamic graph with high precision. On the other hand, we also found that when the compression ratio is lower, TSE performed more favorably. We believe the reasoning behind this symptom is caused by the information loss in DTFT.

# F  STREAMING VS NON-STREAMING

We did our experiments on link prediction in streaming and non-streaming scenarios. In streaming case, we perform inductive learning where the inference is absolutely agnostic of any historical timesteps whereas in streaming case, the historical timesteps is used during training. The result confirmed that the performance of non-streaming cases are generally better than the streaming cases. In both cases, our test performance outperform previous methods.

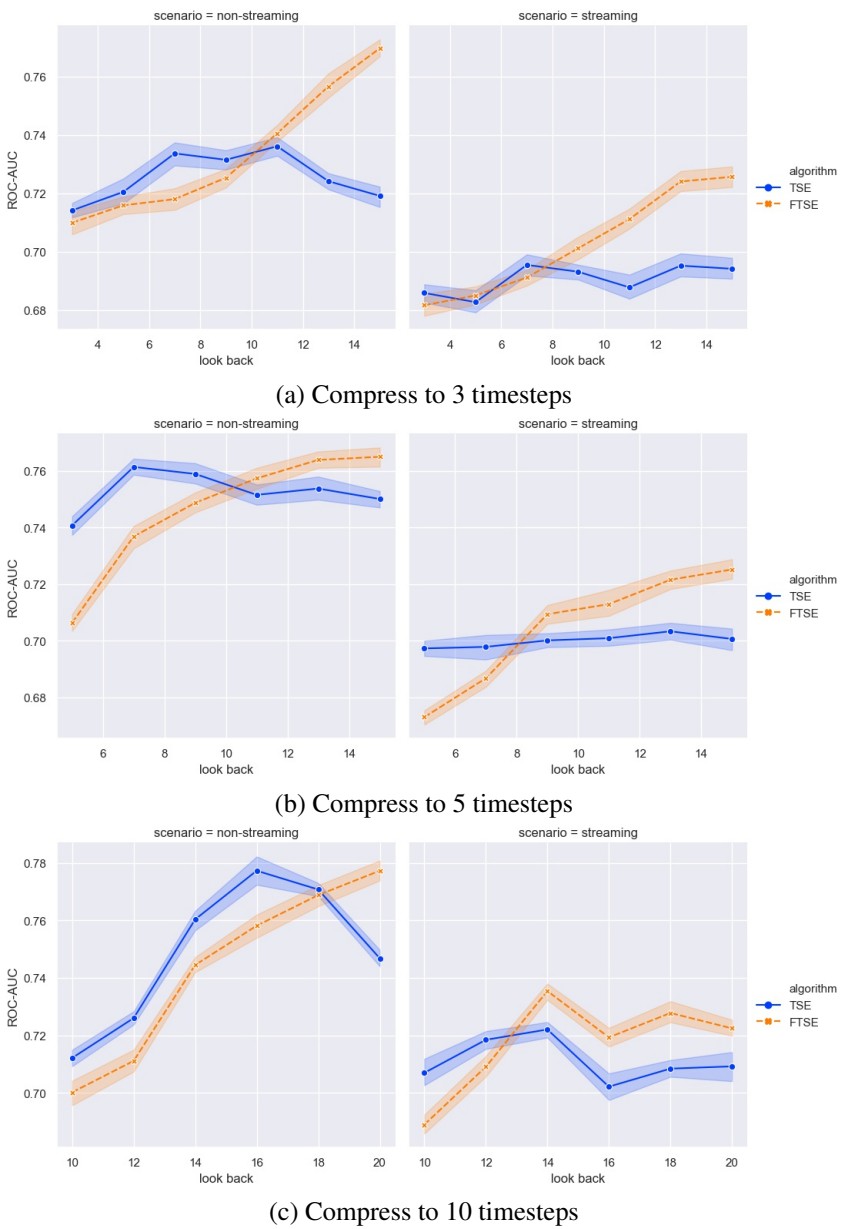

Figure 4: ROC-AUC score on multiple compression ratios where TSE is running with uncompressed data, the result is based on 20 runs and the dataset is UCI, the granularity was also halved compared to the regular settings

| Datasets | UCI | | AS | | SBM | |
|---|---|---|---|---|---|---|
| | **AUC** | **AP** | **AUC** | **AP** | **AUC** | **AP** |
| GCN | 0.464(0.042) | 48(22) | 0.664(0.030) | 314(18) | 0.701(0.004) | 15949(302) |
| GCN+GRU | 0.504(0.022) | 146(13) | 0.802(0.012) | 783(10) | 0.700(0.002) | **16527**(284) |
| GCN+LSTM | 0.508(0.021) | 145(13) | 0.778(0.009) | 1454(12) | 0.697(0.002) | **16549**(278) |
| GAT | 0.579(0.009) | **549**(32) | 0.838(0.016) | 679(30) | 0.650(0.012) | 13398(280) |
| GAT+GRU | 0.506(0.008) | 84(16) | 0.586(0.013) | 223(28) | 0.659(0.007) | 13695(204) |
| GAT+LSTM | 0.504(0.007) | 63(14) | 0.625(0.014) | 303(20) | 0.643(0.008) | 13284(226) |
| EvolveGCN-O | 0.547(0.007) | 447(8) | 0.895(0.006) | **4364**(32) | 0.697(0.005) | 16211(290) |
| EvolveGCN-H | 0.552(0.006) | 532(9) | 0.893(0.008) | **4572**(28) | 0.699(0.004) | 16489(227) |
| TGAT | - | - | - | - | - | - |
| TSE | **0.706**(0.008) | **542**(11) | **0.978**(0.003) | 2894(49) | **0.710**(0.008) | 16202(140) |
| Fourier TSE | **0.704**(0.007) | 528(12) | **0.983**(0.007) | 2987 (66) | **0.708**(0.007) | 16377(167) |

Table 4: The ROC-AUC score (**AUC**) and Average Precision(**AP** at $1e - 6$ scale) on **streaming** link prediction task, mean and standard deviation are reported according to 20 runs of each algorithm

