# OpenReview forum: "Dynamic Graph Representation Learning with Fourier Temporal State Embedding"
_ICLR.cc/2021/Conference — Reject_

### Official Review · AnonReviewer4 · 2020-10-26
**Dynamic graph embedding by Fourier temporal state embedding**

**Rating:** 5
**Confidence:** 4

**Review:**

The paper studies dynamic graph embedding and proposes the method that first transforms the signal of certain edge's existence to its frequency domain by DFT and then considers it as edge features followed by the simplified vanilla GCN.
Pros:
  1. The idea of using DFT to capture the dynamics is simple yet powerful based on the results of experiments.
  2. The experiments are able to show the improvements in various tasks.
Cons:
  1. The definition of the edge existence signal f is not clear (e.g., Eq. 2). It's not defined how to compute the probability in Eq. 2, so I bet the authors directly use existence as Pr=1 and non-existence as Pr=0. But it's not clearly mentioned in Section 4.1. Similarly, how to compute the conditional probability in Eq. 5 is also not clear.
  2. Usually in graph learning (especially GCNs), receptive field denotes the nodes that are used for aggregations. The paper claims the proposed FTSE has a larger receptive field, but didn't give the reason.
  3. The proposed method actually remove the dependencies between two timestamps, which are usually very important information to capture the network dynamics. This is one limitation of the paper.
  4. The notations are inconsistent. In Eq. 6, what is i? In Appendix C, the symbol N is first denoted as the embedding vector dimension, but then used as the number of nodes.
  5. My major concern is the complexity analysis. Based on Eq. 4, the features in frequency domain are computed for every node pairs (i.e., N^2). That's why when conducting the convolutions in Eq. 6, the complexity is O(N^2), which is a bit high compared to the vanilla GCNs where the adjacency matrix is very sparse (i.e., |E| number of edges). Based on this, I think the complexity analysis in Appendix C is not fair. In the analysis, the authors directly assume dense matrix multiplication (e.g., the multiplication between A and X has O(N^3) time complexity), which is true for the proposed method. However, for the baseline which uses vanilla GCN + RNN, GCN only has O(|E|) time complexity.

---

> ### Author Response · Authors · 2020-11-17
> **Thank you for your review and expertise**
>
> Thank you for your review. Your advice and comments on this work help us a lot in the future revision of this paper.
>
> The unclear definition in section 4 would be a key point in our future revision, which needs to be clarified with good details together with the notations in the future revision of this paper.
>
>  The receptive field in this paper corresponds to the number of timestamps times the graph size, which resulted in common confusion and would need to be clarified. We elucidated the reason for using FTSE in section4, where FTSE can embed a very large number of history graph into a fixed-length representation.
>
> The network dynamics are directly captured within the Fourier transform, where all of its information is kept in DTFT from the nature of integral transform.
>
> Your suggestions on GCN's O(|E|) complexity is reasonable for sparse computation. We will clarify that in the revision of this paper. Our method is also O(|E|) in the sense that the DFT will not change the all-zero signals. Then the GCN in our method did not introduce higher complexity.

---

### Official Review · AnonReviewer2 · 2020-10-27
**Temporal sequence is not considered**

**Rating:** 4
**Confidence:** 4

**Review:**


The article presents a new approach for learning representations of dynamic graphs. The method is based on Fourier Transform of edges over time, and separate GCNs are used for each Fourier mode (N FTSE) to stain temporal embeddings. Numerical results illustrate the performance of the method on there graph tasks, namely link prediction, node and edge classifications.

---------------
Strengths:
1. Novel method is presented based on the Fourier Transform of edges over time.
2. Improved performance on some datasets.
3. Complexity analysis is given.
---------------
Weakness:
1. Model violates the temporal sequence.
2. The number of parameters to be learned is large.
3. Experiments are inadequate.

---------------
Details:
I have the following comments about the paper:

1. Temporal sequence violation: The proposed method violates the temporal sequence of the graphs. Computing DFT and the sampling procedure ignores the temporal sequence (time linearity). Note that for the N GCN trained, the adjacency matrices are based on the FTSE which means an embedding at time t, would have information from future time instances.
Intuitively this seems incorrect. The model does not account for the sequential correlation of the graphs over time. I recommend the authors clarify this issue.

2. Number of parameters:  It appears the number of parameters in the proposed model will be significantly more than the compared methods. It appears we will have N different weight matrices to be learned for the N "independent" GCNs for each Fourier mode.  Note that for GCN-RNN based methods such as EvolveGCN, we only have one weight matrix corresponding to the GCN and one for the RNN that are to be learned over time. It appears the model is overparamterized and the comparisons seem unfair.

3. Experiments are inadequate. It is unclear why just three datasets were used for link prediction, and one (different) dataset was used for node and edge classification, respectively. Link prediction and classification are feasible on all five datasets considered. It is unclear why datasets were picked in this manner.
Moreover, the average precision (AP) reported seem very low (scale of 1e-2). This is contrary to the results presented in other paper, where AP are higher, see [3]. At such scale its hard to say one is better than other. For the supervised classification tasks (node and edge), results on just 1 dataset each are given. This is not adequate to make any conclusions.

4. The paper is poorly written, and needs to be checked for language consistency throughout. I recommend having the paper proof-read by a native speaker. E.g., letter
"t" is used for both layers and time instances.

Minor Comment:
Few recent related works are missing:

[1] Sankar, Aravind, et al. "DySAT: Deep Neural Representation Learning on Dynamic Graphs via Self-Attention Networks." Proceedings of the 13th International Conference on Web Search and Data Mining. 2020.

[2] Franco Manessi, Alessandro Rozza, and Mario Manzo.  Dynamic graph convolutional networks. Pattern Recognition, 97, 2020.

[3] Malik, Osman Asif, et al. "Dynamic Graph Convolutional Networks Using the Tensor M-Product." arXiv preprint arXiv:1910.07643 (2020).

---

> ### Author Response · Authors · 2020-11-17
> **Thank you for your review and expert advice**
>
> Thank you very much for your review. It is very helpful for the revision of this paper.
>
> The violation of temporal sequence is the design in this approach. In RNN-based approaches the forward propagation needs to be calculated sequentially, containing multiple convolutions/large matrix multiplication. Our design sees the temporal existence of edges as a graph-structured signal and uses DTFT as an encoding method. The sequential correlation is broken so we will be able to perform Graph Convolution with multiple channels, where each channel corresponded to a Fourier mode. We did not need to use RNNs any more.
>
> For the concern raised to the parameter in the model, we will need to clarify that in the implementation of the method, we used a single GNN weight matrix although using multiple GNNs might even achieve better results.
>
> The difference between experimental matrices comes from the different experimental settings we perform. The baseline results reported in our experiments are coming from the code of EvolveGCN, with the default parameter settings reported by its author. The link prediction task is an imbalanced classification, if all the edges are taken into inference, MAP will normally lower if the inference is based on sampled edges to the best of our knowledge
>
> We are very sorry for the typos and the poor consistency of symbols in the paper, which we are working to revise and polish. We would thank the anonymous reviewer for pointing this out.

---

### Official Review · AnonReviewer1 · 2020-10-28
**More analysis are needed to illustrate the underlying mechanism of the Fourier temporal state embedding**

**Rating:** 4
**Confidence:** 5

**Review:**

The authors propose a temporal embedding approach to handle the timely interactions between nodes. The paper is written clearly for most of the part, despite several typos and grammar mistakes.

The major contribution of the paper is the Fourier temporal state embedding. The authors use it as a plug-in for the standard GNN, so the novelty is limited on the GNN side. The motivations and general solutions for dynamic temporal graph embedding have been intensively mentioned in the recent literature, and the arguments and insights from this paper are relatively standard in this regard.

The highlights of this paper include:
1. a computationally feasible embedding approach that takes account of the temporal information;
2. a discussion on the advantage of the proposed approach in terms of the practical implementation.

Since the contribution of the paper is focused on the Fourier temporal state embedding, my major concerns regarding this approach are summarized as follow.

1. Unclear motivation of the Fourier temporal state embedding. The authors advertise on the advantage of converting the temporal signal to the frequency space throughout the paper, with multiple references on signal processing. However, they provide no evidence of theoretical justifications on:
   a. what are the interpretations and properties of the proposed embedding on the frequency space;
   b. how are they capturing the temporal signals and feeding them to the GNN;
   c. why would the hidden representations (after local aggregation), which reflects the frequency signals according to the construction, possess valid temporal/frequency information after the GCN-type aggregation.
In the absence of the above discussions, even though the numerical results show some promising outcome, it remains mostly unknown why the proposed approach should be considered in the first place.

2. Insufficient comparisons with the previous work of TGAT. The motivation and solution of this paper highly resemble the TGAT proposed by Xu, et al. 2020, from the idea of inventing a temporal embedding functional to converting the signals to the frequency space. Both papers leverage Fourier transformation to some extent, while TGAT has more a clear temporal-kernel interpretation and theoretical guarantee, and the temporal component, which is also reflected via the frequency domain, can be learnt in a data-adaptive fashion.  TGAT also treats the time as a continuous variable, whereas the authors do not study that part of the Fourier transformation in this paper. It seems the idea behind the Fourier temporal state embedding represents a subset of the temporal embedding in TGAT, and the authors do not make clear of the pros and cons compared with the temporal embedding in Xu, et al. 2020.

3. The explanations for the working mechanism are mostly heuristic and not rigorous. The authors mention that the proposed embedding can be viewed as the signal strength under different frequencies.  In this regard, what is the geometric meaning of adding the embedding? Is it still reasonable to apply local convolution with respect to the embedding? How are the frequency components interacting with the features? The authors refer the classical approach from signal processing, but provide no rigorous analysis regarding the consequence of their practice in that regard.

4. A fundamental issue of not treating time as a continuous variable. If the graph structure does not evolve (no nodes and edges are added/deleted), the prediction will be static regardless of the target time by the proposed approach, because the embeddings will not change according to how they are constructed. In many applications, the relative time span, rather than the absolute time value, is more important to reveal the temporal effect. I view this as a methodology limitation that requires improvement.

Based on the above analysis, I believe the presentation and proposals in this version of the paper raise major concerns that eventually outweigh the contributions. Therefore, I vote for rejection.

---

> ### Author Response · Authors · 2020-11-17
> **Thank you very much for the review and expert advice to this work**
>
> Thank you very much for your review of our paper. Your review gave good advice on this work, which is intuitive and helpful for future revision.
>
> The original motivation for using TSE is to see the learning of temporal graph as the learning of a signal pattern in the graph data with the help of signal processing methods. Transforming temporal signals into frequency space is designed to encode the temporal state of edges into a fixed-length representation vector and cut off the sequential dependency in the temporal domain on purpose since the complexity of training an RNN with multiple snapshots of graph features within a certain period is costly in time and memory. One other advantage of using Fourier transform rather than taking directly the temporal signals comes from the fact that the more timesteps we incorporate in our training, generally, the better capacity of our model will be. Integral transform like DTFT transformed the discrete-time signal into a continuous domain, where we can actually form an injective mapping from the temporal domain to the spectral domain with sampling (which is like to formation of DFT).
>
> Our work is different from TGAT in the literature of GNNs. TGAT provides a very intuitive method, constructing an RKHS with DFT. Their method is designed to incorporate time as a feature in the Attention mechanism. Our work directly learns the signal from the graph, which is independent of the actual structure of GNNs.
>
> The rigorousity in mathematical properties of this method is not established formally, which will be one of the important things for our future effort in the revision of this paper.
>
> This method is naturally extended to continuous time cases as DTFT can be generalized to FFT, which naturally deals with signals in the continuous temporal domain. We believe the reason for carrying out experiments with discrete datasets is to be consistent with previous methods.

---

> > ### Comment · AnonReviewer1 · 2020-11-24
> > **Thanks for the response**
> >
> > I want to thank the authors for the response. I decide to maintain the original score.

---

### Official Review · AnonReviewer3 · 2020-10-28
**Official Blind Review #3**

**Rating:** 5
**Confidence:** 5

**Review:**

This paper presents a new method called Fourier temporal state embedding. The motivation for this approach is unclear and should be appropriately justified. In the abstract and introduction, the claim appears to be that previous methods are not time nor memory-efficient, and therefore FTSE is proposed. But this is obviously not true. So it is unclear what this new approach offers compared to the state-of-the-art. In Table 1, why not report performance for more standard baselines like CTDNE? The clarity and writing of this work require significant improvement. There are many incomplete and incorrect sentences throughout the paper that make it difficult if not impossible to understand. In the problem formulation, CTDG and DTDG were originally introduced in the CTDNE paper, but instead a more recent 2020 paper is referenced. Many of the ideas are never fully explained properly. The labels in nearly all the figures need to be appropriately sized, as they are impossible to read.

The motivation of this work needs to be stated clearly. The contribution and differences between existing work need to be clarified and discussed appropriately as well. In addition, some related work on temporal network representation learning is missing and should be appropriately discussed.

Overall, the approach is interesting, the results seem promising, but more work is needed to better position and motivate it. Additional experiments and details would further strengthen it as well.

---

> ### Author Response · Authors · 2020-11-17
> **Thank you for the expert review**
>
> Thank you for your review on our paper. It is true that many unsupervised temporal graph embedding method like CTDNE has been proposed to deal with temporal graph learning tasks, which should be taken into account and credited in this work. We believe those methods are very time-efficient (which requires much less computation than GNNs training).
>
> In this work, we proposed to improve the current state-of-art of current GNN-based methods, which still suffered from high overhead. In the future revision, we will take your advice with extra experiments comparing our method with CTDNE and further discussions comparing our methods with unsupervised counterparts.
>
> The writing will also be carefully revised.

---

### Decision · Program_Chairs · 2021-01-07
**Final Decision**

**Decision:**

Reject

**Comment:**

The paper proposes Fourier temporal state embedding, a new technique to embed dynamic graphs.  However, the paper needs to be improved in writing, computational complexity analysis, and more thorough baseline comparisons.